# DUAL PROMPT TUNING FOR DOMAIN-AWARE FEDERATED LEARNING

## ABSTRACT

Federated learning is a distributed machine learning paradigm that allows multiple clients to collaboratively train a shared model with their local data. Nonetheless, conventional federated learning algorithms often struggle to generalize well due to the ubiquitous domain shift across clients. In this work, we consider a challenging yet realistic federated learning scenario where the training data of each client originates from different domains. We address the challenges of domain shift by leveraging the technique of prompt learning, and propose a novel method called Federated Dual Prompt Tuning (Fed-DPT). Specifically, Fed-DPT employs a pre-trained vision-language model and then applies both visual and textual prompt tuning to facilitate domain adaptation over decentralized data. Extensive experiments of Fed-DPT demonstrate its significant effectiveness in domain-aware federated learning. With a pre-trained CLIP model (ViT-Base as image encoder), the proposed Fed-DPT attains 68.4% average accuracy over six domains in the DomainNet dataset, which improves the original CLIP by a large margin of 14.8%.

## 1 INTRODUCTION

Federated Learning (FL) is a privacy-preserving machine learning technique that allows training a centralized model across decentralized devices while keeping data localized. The general federated learning paradigm involves many rounds of local training and global parameter aggregation(McMahan et al., 2017), which enables learning from decentralized data but is susceptible to two primary challenges: extensive domain shift across clients and limited communication efficiency.

As federated learning assumes that each local dataset is independently collected, the data is inevitably heterogeneous across clients. This heterogeneity highly challenges FL algorithms and results in a considerable performance gap realative to centralized training. In practice, as it is difficult to collect a real federated dataset from multiple clients, prior works choose to partition one single dataset into several splits with non-independently and identically distributed (non-i.i.d.) labels so that to simulate the desired heterogeneity. However, we argue that this setting overlooks a crucial characteristic of federated datasets, which is that data from various clients may originate from different domains, and the data should exhibit significant differences in the input space, rather than simple non-i.i.d. characteristics based on labels.

Therefore, we consider a more challenging yet realistic scenario: the clients desire to deal with the same machine learning problem (e.g., image classification with the same target categories), yet their local data originate from different domains. Following the practice of prior works (Peng et al., 2020), we formulate this scenario using domain-aware datasets like DomainNet (Peng et al., 2019), where there are labeled images sourced from six distinct domains with quite different styles such as real-world, paining, and sketch. Due to the large diversity in input, conventional domain-agnostic federated learning approaches often struggle to generalize well in this problem.

The extensive domain shift and data heterogeneity also challenge the convergence abilities of decentralized training, leading to non-robust federated learning models. This effect becomes more pronounced when dealing with larger models. For example, it is observed that in some federated learning scenarios with highly heterogeneous data, a 152-layer ResNet (He et al., 2016) with 60M parameters even performs worse than a 50-layer ResNet with 26M parameters (Qu et al., 2022). A potential way to tackle this challenge is leveraging parameter-efficient training strategies, which employ a pre-trained model and only fine-tune a small portion of parameters (Lu et al., 2023).

With the advancement of Contrastive Language-Image Pre-training (CLIP) models (Radford et al., 2021), it is very convenient to develop parameter-efficient learning protocols by prompt tuning techniques. Specifically, these methods employ a pre-trained CLIP model, freezing both its image and text encoders, and feed it with learnable tokens (i.e., prompts) attached to the original input (Zhou et al., 2021; Jia et al., 2022). By optimizing only the prompt tokens, the model can quickly adapt to downstream datasets and domains.

In this work, we propose **Fed**erated **D**ual **P**rompt **T**uning (**Fed-DPT**), a novel federated learning approach to overcome the challenges mentioned above. In detail, we address the challenge of parameter-efficiency by harnessing the techniques of CLIP and prompt learning for both visual and textual inputs, making our method friendly to communication cost and robust to federated optimization with heterogeneous data. While existing CLIP-based methods share one single prompt learner (Guo et al., 2023) or adaptor (Lu et al., 2023) with all images from various domains, we further tackle the challenge of domain shift across clients by introducing domain-specific prompts and coupling visual and textual representations by self-attention, leading to a domain-aware federated learning method. In scenarios where each domain possesses substantial data with a distinct statistical distribution, our approach is particularly suited to a cross-silo federated learning context.

Extensive experiments demonstrate the significant effectiveness of our method. Remarkably, we obtain a 68.4% average accuracy over six domains in the DomainNet dataset, outperforming the original CLIP model by 14.8%. Compared with conventional federated learning methods such as FedAvg (McMahan et al., 2017) and FedProx Li et al. (2020b), and existing domain-agnostic CLIP-based approaches such as PromptFL (Guo et al., 2023) and FedCLIP (Lu et al., 2023), our Fed-DPT consistently achieves superior performance on three benchmarks.

## 2 RELATED WORK

**Federated learning.** The concept of federated learning was first introduced in the Federated Averaging (FedAvg) paper (McMahan et al., 2017). to address machine learning problems with massively distributed private data. To enhance the learning potential of FedAvg, FedProx (Li et al., 2020b) introduces a $\ell_2$ regularization term into the original federated learning objective. In addition, based on the success of FedAvg, many follow-up works improve federated learning in terms of privacy-preserving potentials (Wei et al., 2020; Truex et al., 2019), robustness to heterogeneous data (Karimireddy et al., 2020; Li et al., 2019), communication efficiency (Konečný et al., 2016; Sattler et al., 2019), and compatibility to model architectures (Li et al., 2020a; Qu et al., 2022). In contrast to general federated learning methods that simulate non-i.i.d. data by partitioning datasets in the label space, many recent works consider federated learning in a more realistic context of domain adaptation (Yao et al., 2022; Shenaj et al., 2023; Peng et al., 2020). Recently, based on the advances in multi-modal contrastive learning (Radford et al., 2021), various works develop CLIP-based federated learning methods. For example, FedCLIP (Lu et al., 2023) uses a pre-trained CLIP model and performs federated training on an additional adaptor layer, and PromptFL (Guo et al., 2023) proposes to use prompt learning methods for federated optimization.

**Vision-language models.** Following the success of contrastive pre-training in visual modality (He et al., 2020; Chen et al., 2020; Grill et al., 2020; Caron et al., 2021; Chen & He, 2021; Chen et al., 2021), multi-modal contrastive pre-training has become a common paradigm in recent years as well. A representative work is CLIP (Radford et al., 2021), which jointly pre-trains a visual and a textual encoder using an InfoNCE objective (Gutmann & Hyvärinen, 2010) with around 400 million curated image-text pairs. ALIGN (Jia et al., 2021) improves CLIP by scaling up the training dataset to 1.8 billion noisy image-text pairs, and BASIC (Pham et al., 2021) further increases the scale of both data and model. As a result, such CLIP-like models allow zero-shot inference when it comes to transfer learning on downstream tasks.

**Prompt tuning.** While fine-tuning a pre-trained model for downstream machine learning tasks has traditionally dominated the field of transfer learning, recent progress in prompt learning offers a compelling alternative. Specifically, the prompt tuning techniques fine-tune learnable prompt tokens attached to CLIP's inputs instead of training the entire model (Zhou et al., 2021; 2022; Wang et al., 2023; Yao et al., 2023). There also exist prompt tuning protocols for visual modality (Jia et al., 2022) and both visual and textual modalities (Yao et al., 2021; Zang et al., 2022). Similarly, there

are adapter-based methods designed for CLIP-like models, which also freeze the encoders and only fine-tune several newly attached layers on top of them (Gao et al., 2021; Zhang et al., 2021).

## 3 Preliminaries

### 3.1 Contrastive Language-Image Models

Contrastive Language-Image Pre-training (Radford et al., 2021) is a weakly supervised learning paradigm that combines visual and language encoders to solve image recognition problems. Formally, CLIP has an image encoder $\boldsymbol{F}_V : \mathbb{R}^{3 \times w \times h} \to \mathbb{R}^d$ where $w$ and $h$ denotes the input image's spatial resolution and $d$ denotes the dimension of the latent space, and a text encoder $\boldsymbol{F}_T : \mathbb{R}^{l \times d_e} \to \mathbb{R}^d$ where $l$ is the length of input sentence and $d_e$ is the dimension of word embedding (512 for CLIP's transformer). CLIP is trained by image-text pairs, in which the text briefly describes the information in the image. By encoding both the image and text into the same latent space, CLIP can learn an alignment between visual and textual input with a contrastive loss (Gutmann & Hyvärinen, 2010).

The CLIP-like vision-language models are generally pre-trained on web-scale datasets with several hundred million image-text pairs (Radford et al., 2021; Jia et al., 2021; Pham et al., 2021). As a result, these models excel in recognizing image features and support zero-shot inference by aligning visual features to text queries. Specifically, given an image, the encoder $\boldsymbol{F}_V$ maps it into a vectorized feature $\boldsymbol{f}_V \in \mathbb{R}^d$, and given several text queries (i.e., class names) such as "cat", "dog", and "horse", the encoder $\boldsymbol{F}_T$ maps each of them into a vector $\boldsymbol{f}_T \in \mathbb{R}^d$ with a prompt. By computing the cosine similarity between visual and textual features, CLIP classifies the image into the $i$-th class with probability

$$p_i = \frac{\exp(<\boldsymbol{f}_V, \boldsymbol{f}_T^i> /\tau)}{\sum_j \exp(<\boldsymbol{f}_V, \boldsymbol{f}_T^j> /\tau)}, \tag{1}$$

where $< \cdot, \cdot >$ denotes dot product, $\tau$ denotes a temperature coefficient, and note that both $\boldsymbol{f}_V$ and $\boldsymbol{f}_T$ have been $\ell_2$ normalized.

### 3.2 Prompt Tuning for Vision and Language

Despite CLIP's impressive zero-shot inference capabilities, it still exhibits a noticeable accuracy gap in comparison to in-domain fine-tuning. However, tuning CLIP's model parameters can easily break the well-established alignment between vision and language, and CLIP therefore loses the ability of open-vocabulary inference as well as address the challenge of domain adaptation. Instead, prompt tuning attaches learnable tokens to the input, leaving the feature encoders fixed. This approach allows the model to retain its zero-shot and open-set inference abilities while significantly improving in-domain accuracy.

**Textual Prompt Tuning (TPT)**. As previously mentioned, CLIP's text query consists of a hand-crafted prompt (also referred to as prefix) such as "A photo of a" and a class name such as "dog". TPT replaces the prefix by learnable vectors (Zhou et al., 2021). Formally, it feeds the text encoder $\boldsymbol{F}_T$ with a sequence of trainable vectors $\boldsymbol{t}_1$, $\boldsymbol{t}_2$, ..., $\boldsymbol{t}_m$ followed by the embedding of class name, where each $\boldsymbol{t} \in \mathbb{R}^{d_T}$ represents a "latent word" that has the same dimension as CLIP's word embedding. During training, both CLIP's vision and language encoders are frozen and only the prompt vectors $\boldsymbol{t}_1$, $\boldsymbol{t}_2$, ... $\boldsymbol{t}_m$ are optimized.

**Visual Prompt Tuning (VPT)**. The prompt tuning protocol also works for visual input if the image encoder is a transformer-like model such as the Vision Transformer (Dosovitskiy et al., 2021). Specifically, this method attaches $n$ trainable vectors $\boldsymbol{i}_1$, $\boldsymbol{i}_2$, ..., $\boldsymbol{i}_n$ to the patch-wise embedded image, and uses an additional head to project the output, where each prompt token $\boldsymbol{i} \in \mathbb{R}^{d_I}$ has the same dimension as image embeddings (e.g., 768 for ViT-Base). In VPT, only the prompt tokens and the head are optimized.

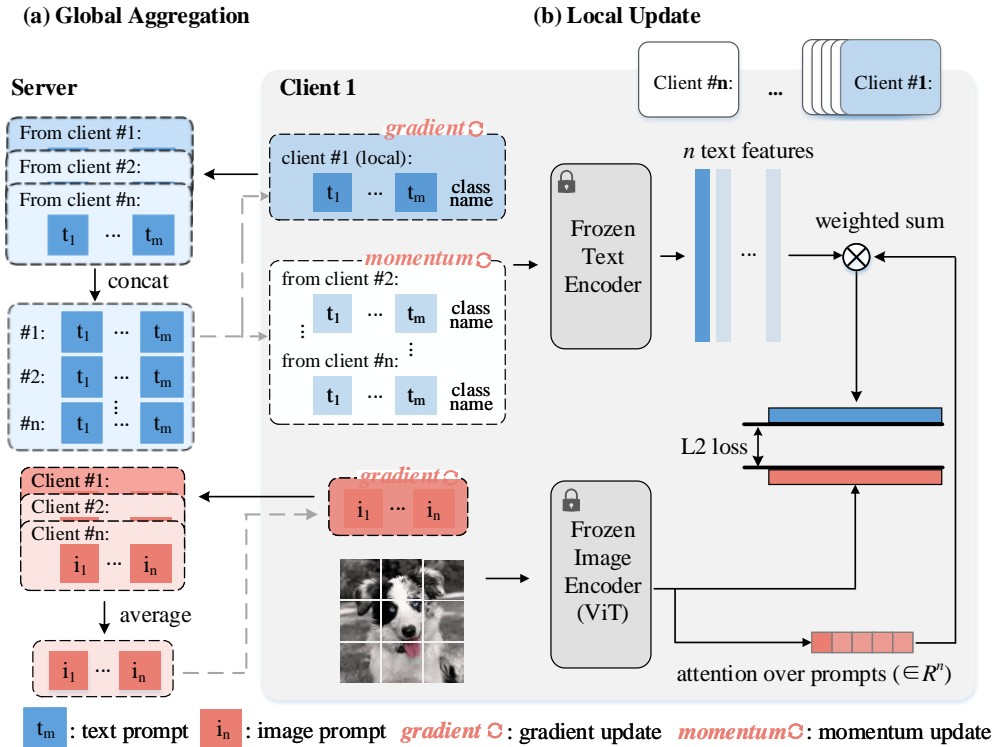

**(a) Global Aggregation**

**(b) Local Update**

$t_m$ : text prompt    $i_n$ : image prompt    *gradient* ↻ : gradient update    *momentum* ↻ : momentum update

Figure 1: Overview of Fed-DPT framework. (a) *Global Aggregation* is the pipeline of Parameter aggregation. We aggregate textual prompts by concatenating the domain-specific tokens from each client, and aggregate visual prompts by averaging. (b) *Local Update* is our Local training framework. For each client, we feed the text encoder with $n$ text prompts followed by class names, where one is optimized by the gradients and the rest $n-1$ are loaded from other clients with momentum update. We feed the image encoder with $n$ learnable prompt tokens followed by patch-wise embedded images, where the prompt tokens are optimized by gradients.

# 4 METHODOLOGY

We present **Fed**erated **D**ual **P**rompt **T**uning (**Fed-DPT**), a novel federated learning method to overcome the challenges of domain shift and parameter efficiency. We detail our method below.

## 4.1 PROBLEM FORMULATION

There are $n$ clients that desire to deal with the same machine learning problem, e.g., image classification with the same target categories. The $n$ clients possess their own training data that originate from $n$ distinct domains. In other words, each client stands for a specific domain. We simulate this scenario using domain adaptation datasets like DomainNet (Peng et al., 2019), which encompass images from six different domains including clipart, information graph, painting, quickdraw, real-world images, and sketch. As the image features exhibit significant variation across different domains, it is indeed a challenging task for federated optimization. However, it is a realistic scenario because many times, the data heterogeneity between clients arises from differences in feature distributions rather than label distributions. Notably, our setting is compatible with the task that clients have non-i.i.d. labels. In our ablation study, we also further divide each domain into five splits with non-i.i.d. categories (see Section 5.3 for details).

## 4.2 LOCAL TRAINING

With CLIP, a very simple way to deal with domain shift is to use domain-aware prompt contexts for text queries. For example, in DomianNet, when we use prefix "a painting of a" for the painting

domain, and use "a sketch of a" for the sketch domain, the predictions can be more accurate and robust. This idea is also referred to as domain-specific prompts (Ge et al., 2022), while employing learnable text prompts can further improve the predictive performance. Inspired by this observation, we propose to use domain-specific prompts for CLIP's text encoder. Formally, we define a text prompt by a sequence of learnable tokens:

$$\boldsymbol{P}_T = [\boldsymbol{t}]_1[\boldsymbol{t}]_2 \ldots [\boldsymbol{t}]_m \in \mathbb{R}^{m \times d_e}, \tag{2}$$

where $m$ is the length of prompt and each token $[\boldsymbol{t}]_i \in \mathbb{R}^{d_e}$ has the same dimension as CLIP's word embedding.

Figure 1 illustrates our Fed-DPT's local training framework. We initialize Fed-DPT by loading the same CLIP model for each client and freezing the parameters of both the image encoder $\boldsymbol{F}_V$ and the text encoder $\boldsymbol{F}_T$. For our task, we have $n$ text prompts $\boldsymbol{P}_T^1, \boldsymbol{P}_T^2, \ldots, \boldsymbol{P}_T^n$ corresponding to the $n$ domains. During local training, the $n$ text prompts are shared among the clients, yet the $i$-th prompt $\boldsymbol{P}_T^i$ can only be trained by the $i$-th client (we will detail this mechanism later). We separately feed the encoder $\boldsymbol{F}_T$ with all the $n$ text prompts followed by a class name, leading to $n$ representation vectors $\boldsymbol{f}_T^1, \boldsymbol{f}_T^2, \ldots, \boldsymbol{f}_T^n$, where

$$\boldsymbol{f}_T^i = \boldsymbol{F}_T(\boldsymbol{P}_T^i, [\text{class name}]). \tag{3}$$

Note that we suppose each $\boldsymbol{f}_T^i$ stands for the representation of the class name in the $i$-th domain.

We define visual prompts by $n$ learnable tokens $[\boldsymbol{v}]_1, [\boldsymbol{v}]_2, \ldots, [\boldsymbol{v}]_n$ which also correspond to the $n$ domains. During local training, we feed the visual encoder $\boldsymbol{F}_V$ (ViT architecture) with a class token [cls] (directly loaded from CLIP), $n$ visual prompts, and the patch-wise embedded image, leading to an image representation vector

$$\boldsymbol{f}_V = \boldsymbol{F}_V([\text{cls}], [\boldsymbol{v}]_1, [\boldsymbol{v}]_2, \ldots, [\boldsymbol{v}]_n, [\text{image}]). \tag{4}$$

Beside of $\boldsymbol{f}_V$, we meanwhile obtain attention scores between the [cls] token and visual prompts. Formally, denoting $\boldsymbol{q}_{\text{cls}}$ as the query vector of the class token, and $\boldsymbol{k}_i$ as the key vector of the $i$-th prompt token in $\boldsymbol{F}_V$'s last self-attention block, we have $\boldsymbol{w} = [w_1, w_2, \ldots, w_n]$ with

$$\boldsymbol{w}_i = \frac{\exp(<\boldsymbol{q}_{\text{cls}}, \boldsymbol{k}_i > /\tau_d)}{\sum_j \exp(<\boldsymbol{q}_{\text{cls}}, \boldsymbol{k}_j > /\tau_d)}, \tag{5}$$

where $\tau_d$ is a temperature coefficient. We regard each component $w_i$ as the visual feature's correlation to the $i$-th domain, and compute the final text output by

$$\boldsymbol{f}_T = \sum_{i=1}^n w_i \boldsymbol{f}_T^i. \tag{6}$$

During the local training process in the $i$-th client, we optimize the $i$-th text prompt $\boldsymbol{P}_T^i$ and all the $n$ visual prompts $[\boldsymbol{v}]_1, [\boldsymbol{v}]_2, \ldots, [\boldsymbol{v}]_n$ by a $\ell_2$ loss applied to $\boldsymbol{f}_V$ and $\boldsymbol{f}_T$:

$$\mathcal{L} = <\boldsymbol{f}_V, \boldsymbol{f}_T > /||\boldsymbol{f}_V|| \cdot ||\boldsymbol{f}_T||. \tag{7}$$

Here we explain why we optimize these parameters. We desire the $i$-th text prompt $\boldsymbol{P}_T^i$ to represent the features of the $i$-th domain in the latent space of textual embeddings. However, the $i$-th client only possesses images from the $i$-th domain, so we cannot train $\boldsymbol{P}_T^j$ ($j \neq i$) yet instead load them from other clients. We introduce visual prompts to detect the correlations between an input image and the $n$ domains, so it is fine to optimize all of them. A detailed comparison of different training strategies can be found in our ablation study (see Table 4a and 4b for details). Also note that in our experiments, we find the $\ell_2$ loss yields better predictive performance and allows more flexible training compared with cross-entropy, while we do not observe any collapse issues.

## 4.3 PARAMETERS AGGREGATION

As mentioned above, for the $i$-th client, we optimize $\boldsymbol{P}_T^i$ by gradients and load $\boldsymbol{P}_T^j$ ($j \neq i$) from other clients, so the aggregation of text prompts does not involve parameter merging processes (e.g. averaging). We illustrate the aggregation pipeline in Figure 1. For easy understanding, we suppose there is a centralized parameter server — actually, Fed-DPT also works for decentralized

communication — and the clients upload their corresponded text prompt to it in each communication round. The server concatenates the $n$ uploaded text prompts and then sends to every client. For visual parameters, as all visual prompts are optimized by every client, we perform federated averaging in the server and then send the merged parameters to each client. Note that we do not need to share CLIP encoders' parameters as each client is initialized with the same CLIP model and its parameters are frozen during training.

This parameter aggregation paradigm works well for Fed-DPT, yet may create a minor problem for the text encoder. Specifically, after each communication round, the external text prompts of the $i$-th client, i.e., $\boldsymbol{P}_T^j$ ($j \neq i$) will be re-loaded. We observe that this sudden change of parameters often negatively affects our model. To address this issue, we propose to apply momentum update (also referred to as exponential moving average) to the external text prompts. Formally, we have

$$[\boldsymbol{t}]^s = \alpha[\boldsymbol{t}]^{s-1} + (1 - \alpha)[\boldsymbol{t}], \tag{8}$$

where $[\boldsymbol{t}]^s$, $[\boldsymbol{t}]^{s-1}$ denote the prompt tokens at the $s$ and $s - 1$ step, and $[\boldsymbol{t}]$ denotes the vector received from other clients, and $\alpha \in [0, 1]$ is a coefficient to control the smoothness. The details of our ablation study related to momentum update can be found in Table 4a.

### 4.4 Discussion of Privacy

During the process of global aggregation, our approach involves providing each client with text prompts from every other client. However, in prompt learning techniques, these prompt tokens are not considered as data but a part of model parameters since they are learnable and updated by local gradients. Thus, our method with shared prompts has the same level of privacy-preserving capabilities to FedAvg which shares model parameters globally.

As for text prompts, it is actually very difficult to decode them back to meaningful words in natural language because these tokens are trained in a continuous space and there local optima are not necessarily associated to specific natural words. Here we follow CoOp (Zhou et al., 2021) to decode each text prompt by finding a standard vocabulary word with minimum Euclidean distance to it in the embedding space, and summarize the interpretation results for DomainNet in Table 5.

## 5 Experiments

### 5.1 Experimental Setup

**Datasets**. We evaluate our Fed-DPT and baseline methods on the following three domain adaptation image classification benchmarks:

- DomainNet (Peng et al., 2019). The DomainNet dataset has around 600,000 images spanning 345 categories from six domains, which covers diverse image styles including clipart, infograph, painting, quickdraw, real, and sketch.
- OfficeHome (Venkateswara et al., 2017). The OfficeHome dataset consists of approximately 15,500 images depicting everyday objects in 65 classes. It further categorizes the images into four domains: art, clipart, product, and real-world.
- PACS (Li et al., 2017). The PACS dataset contains around 10,000 images drawn from seven categories and four domains, including photo, sketch, cartoon, and painting styles.

**Baseline methods**. We first consider the baselines of CLIP and its adapted models to federated learning. The *Zero-shot CLIP*, which infers by aligning images to their class names with a hand-crafted prompt, is a direct baseline to evaluate whether in-domain tuning is necessary for vision-language models in federated learning. We also introduce *Single-domain tuning*, which applies textual prompt tuning (Zhou et al., 2021) to CLIP only in the local domain, as another baseline to testify whether it is helpful to combine the information across multiple domains. There are also domain-agnostic federated learning approaches based on CLIP such as *PromptFL* (Guo et al., 2023) and *FedCLIP* (Lu et al., 2023), which train text prompt and an adapter layer in federated learning fashion, respectively. To further validate the effectiveness of our method, we also compare it with conventional federated learning algorithms *FedAvg* (McMahan et al., 2017) and *FedProx* (Li et al.,

Table 1: Test accuracy (%) on **DomainNet**. The *info g.*, *paint.*, and *quick d.* denote the domains of *infogragh*, *painting*, and *quickdraw*, respectively. Our results are marked in  blue . The best results in each domain are **bolded**.

| Method | DomainNet | | | | | | |
|---|---|---|---|---|---|---|---|
| | clipart | info g. | paint. | quick d. | real | sketch | avg. |
| Z. S. CLIP (Radford et al., 2021) | 66.1 | 40.6 | 62.3 | 13.5 | 80.4 | 58.5 | 53.6 |
| Single-Domain Tuning | 72.3 | 47.2 | 67.1 | 18.8 | 83.6 | 65.8 | 59.1 |
| *Conventional federated learning methods:* | | | | | | | |
| FedAvg (*ResNet-50 backbone*) | 40.2 | 61.1 | 57.6 | 33.5 | 75.6 | 60.3 | 54.7 |
| FedAvg (*ViT-B/16 backbone*) | 42.4 | 60.7 | 57.0 | 30.4 | 79.8 | 61.1 | 55.2 |
| FedProx (*R-50*) (Li et al., 2020b) | 41.5 | 62.0 | 56.8 | 34.9 | 79.2 | 62.6 | 56.2 |
| FedProx (*ViT-B*) (Li et al., 2020b) | 40.5 | **63.1** | 57.4 | 29.7 | 81.2 | 59.8 | 55.3 |
| *Domain-agnostic vision-language tuning methods:* | | | | | | | |
| PromptFL (Guo et al., 2023) | 76.0 | 50.2 | 70.4 | 33.5 | 81.2 | 67.8 | 63.2 |
| FedCLIP (Lu et al., 2023) | 74.1 | 48.3 | 68.5 | 31.8 | 80.5 | 58.6 | 60.3 |
| Fed-DPT (ours) | **77.5** | **63.1** | **70.5** | **41.6** | **85.7** | **72.1** | **68.4** |

2020b) that are not based on CLIP. We equip these two baselines by a 50-layer ResNet (He et al., 2016) and a base-scale vision transformer with 16×16 patch size (Dosovitskiy et al., 2021), both being pre-trained on ImageNet-1k (Deng et al., 2009).

**Implementation details**. For our Fed-DPT, we employ a pre-trained CLIP model with a ViT-Base/16 image encoder, so each textual and visual prompt token has the dimension of 512 and 768, respectively. We set the length of each textual prompt sequence $m = 16$ for better robustness, which follows the practice of TPT (Zhou et al., 2021). By default, the number of clients is determined by the number of domains for each dataset, i.e. $n = 6$ for DomainNet and $n = 4$ for OfficeHome and PACS. We train both our model and the baseline models for 200 epochs and execute the aggregation or broadcast process after every one epoch. We train the ResNet-based models and prompt tokens by a SGD optimizer with 0.01 learning rate, 0.9 momentum, and 0.005 weight decay. Fed-DPT instead uses AdamW (Loshchilov & Hutter, 2019) optimizer with $\beta_1 = 0.9$, $\beta_2 = 0.999$, 5e-4 learning rate, and 0.01 weight decay for transformer-based models. We set the temperature coefficient $\tau_d = 0.1$ in Equation 5, and set the momentum update ratio $\alpha = 0.99$ in Equation 8. If not specified, all reported results are average numbers over three trials.

## 5.2 MAIN RESULTS

Table 1 shows the evaluation results of our method and baselines on DomainNet. We observe that our Fed-DPT outperforms the baseline methods by a large margin in terms of average accuracy over six domains. Notably, while the zero-shot CLIP and single-domain tuning protocols fail to obtain reasonable accuracy in the "quickdraw" domain, our Fed-DPT improves this number to 41.6%, which empirically validates the effectiveness of our approach. Benefiting from the technique of prompt learning that introduces very small number of trainable parameters, we find our Fed-DPT to perform very robust to big models. In contrast, conventional methods such as FedAvg (McMahan et al., 2017) and FedProx (Li et al., 2020b) only yield very marginal improvements, or even incur performance degradation when changing the backbone from ResNet-50 (26M parameters) to ViT-Base (86M parameters). Compared to the domain-agnostic prompt learning methods, our Fed-DPT attains higher average accuracy and lower standard deviation (13.8% vs. 16.5% for FedCLIP and 16.4% for PromptFL). This is possibly because our method considers each image's feature representation in all domains, which makes our predictions more robust to domain shift.

We further evaluate the models on the other two benchmarks and summarize the results in Table 2. The experiments on OfficeHome and PACS also support our conclusion of Fed-DPT's effectiveness by demonstrating higher average accuracy and lower deviation across domains. Specifically, we improve the zero-shot CLIP by 4.3% average accuracy and 0.3% standard deviation over four domains in OfficeHome. We also observe that overall, the prompt-based methods consistently outperform

Table 2: Test accuracy (%) on **OfficeHome** and **PACS**. Domains include *art*, *clipart*, *product*, and *real-world* for OfficeHome, and *photo*, *art painting*, *cartoon*, and *sketch* for PACS. Our results are marked in blue . The best results in each domain are **bolded**.

| Method | OfficeHome | | | | | PACS | | | | |
|---|---|---|---|---|---|---|---|---|---|---|
| | Ar | Cl | Pr | Rw | Avg. | P | A | C | S | Avg. |
| Zero-Shot CLIP | 79.5 | 63.1 | 85.3 | 86.5 | 78.6 | 99.8 | 96.9 | 98.8 | 87.7 | 95.8 |
| Single-Domain | 80.0 | 65.2 | 87.5 | 86.9 | 79.9 | 99.8 | 97.2 | 99.1 | 88.9 | 96.3 |
| *Conventional federated learning methods:* | | | | | | | | | | |
| FedAvg (*ResNet-50*) | 66.3 | 49.4 | 77.1 | 77.9 | 67.7 | 89.6 | 52.5 | 78.6 | 76.1 | 74.2 |
| FedAvg (*ViT-B/16*) | 67.9 | 49.6 | 77.5 | 81.0 | 69.0 | 91.3 | 54.8 | 79.2 | 77.9 | 75.8 |
| FedProx (*ResNet-50*) | 68.8 | 50.5 | 78.6 | 80.3 | 69.6 | 91.7 | 57.0 | 81.8 | 80.2 | 77.7 |
| FedProx (*ViT-B/16*) | 70.4 | 51.3 | 80.3 | 82.4 | 71.1 | 92.0 | 59.4 | 83.5 | 81.6 | 79.1 |
| *Domain-agnostic vision-language tuning methods:* | | | | | | | | | | |
| PromptFL | 79.8 | 65.6 | 89.5 | 89.1 | 81.0 | 99.9 | 97.1 | 99.0 | 90.6 | 96.7 |
| FedCLIP | 79.1 | 65.0 | 88.6 | 88.4 | 80.3 | 99.8 | 97.4 | 98.9 | 89.0 | 96.3 |
| Ours | **82.6** | **68.2** | **90.5** | **90.3** | **82.9** | **99.9** | **98.0** | **99.1** | **91.7** | **97.2** |

the conventional federated learning algorithms that require to train the entire model. This confirms the benefits of employing parameter-efficient approaches in federated learning, and explains why we choose to use prompt tuning to address the domain shift issues.

### 5.3 Ablation Studies

**Model components.** We first dissect our Fed-DPT model to ablate its performance gains. Overall, Fed-DPT comprises two primary components: visual prompts and domain-specific text prompts. By dissecting these components, we get three more variants of our method: 1) *Visual Only*, it leverages learnable prompt tokens for only image input and uses CLIP's hand-crafted prompt for texts. 2) *Textual Only*, it discards the visual prompt tokens of Fed-DPT and uses learnable text prompts only. Note that in the absence of visual prompts, we cannot get the weight $w_i$ (see Equation 5 and 6) for each domain, so the text prompts from external clients should also be discarded. We instead aggregate the textual prompts by federated averaging (McMahan et al., 2017). 3) *Domain-Agnostic DPT*, it retains both Fed-DPT's visual and textual prompts but decouples them, i.e., we do not perform the weighted sum process in Equation 6, which can be considered as a simple combination of the modes *Textual Only* and *Visual Only*.

We summarize the results in Table 3. Since we introduce visual prompt tuning for combining domain information rather than enhancing the visual feature extraction abilities, we do not attach an additional head for the image encoder as in (Jia et al., 2022). Therefore, the *Visual Only* mode cannot yield significant performance improvements. We also observe that tuning textual prompts results in a 5.5% increase in accuracy, and when tuning them in a federated learning fashion, we achieve an additional 4.1% improvement (*Textual Only*). Notably, compared to the simple visual-and-textual prompt tuning with 63.5% accuracy, our Fed-DPT achieves a much higher result of 68.4%, which demonstrates the crucial significance of our domain-aware mechanism.

**Momentum update, prompt length, and communication frequency.** In addition, we consider three more factors that may affect our results. As mentioned in Section 4, we update the external text prompts by exponential moving average to prevent parameters' sudden change. Here we present comparisons regarding the update mechanism for text prompts in Table 4a, where the accuracy drops by 2.2% in the absence of momentum update. If we train all text prompt tokens in every client, i.e., we disregard the relationship between text prompts and domains, the accuracy drops by 4.4% as it makes Fed-DPT a domain-agnostic approach.

By default, we aggregate the visual prompt tokens by federated averaging, as separately training each token in a specific domain does not yield better performance (see Table 4b). As shown in Table 4c, we set the length of each textual prompt sequence to $m = 16$, as it works more robust than a shorter prompt ($m = 4$), and when we further increase the length, the model tends to overfit and

Table 3: Ablation study to model components. We report the average accuracy (%) over six domains in DomainNet. The *v. prompt* and *t. prompt* denote whether using visual or textual prompts. Our default setup is marked in blue .

| Method | federated | v. prompt | t. prompt | domain-specific | acc. |
|---|---|---|---|---|---|
| Zero-Shot CLIP | ✗ | ✗ | ✗ | ✗ | 53.6 |
| Single-Domain Tuning | ✗ | ✗ | ✓ | ✗ | 59.1 (+5.5) |
| Visual Only | ✓ | ✓ | ✗ | ✗ | 54.2 (+0.6) |
| Textual Only | ✓ | ✗ | ✓ | ✗ | 63.2 (+9.6) |
| Domain-Agnostic DPT | ✓ | ✓ | ✓ | ✗ | 63.5 (+9.9) |
| Fed-DPT | ✓ | ✓ | ✓ | ✓ | 68.4 (+14.8) |

accuracy drops. In Table 4d we also assess the impact of communication frequency by varying it to 0.5, 1, and 2 training epochs per communication round. It shows that compared to our default setup of one epoch per communication round, more frequent aggregation (0.5 epoch/round) does not lead to improved performance, while conversely, infrequent communication (2 epochs/round) results in a 0.5% accuracy degradation.

Table 4: Ablation studies. We report the average accuracy over six domains in DomainNet. The *mtm.* denotes momentum update. Our default setup is marked in blue . The best results of each ablation study is **bolded.**

(a) Text prompt update.

| Mode | acc. |
|---|---|
| w/ mtm. | **68.4** |
| w/o mtm. | 66.2 |
| train all | 64.0 |

(b) Visual prompt update.

| Mode | acc. |
|---|---|
| average | **68.4** |
| split w/ mtm. | 68.3 |
| split w/o mtm. | 67.5 |

(c) Prompt length.

| #tokens | acc. |
|---|---|
| 4 | 67.5 |
| 16 | **68.4** |
| 32 | 68.0 |

(d) Comm. frequency

| #eps/round | acc. |
|---|---|
| 0.5 | **68.4** |
| 1 | **68.4** |
| 2 | 67.9 |

**Decentralization**. By default, we consider each domain in the dataset as a single client, leading to non-identical feature distributions yet the same class distribution across clients. To further testify our method's effectiveness and flexibility, we conduct a more challenging scenario on DomainNet where each domain is further divided into five clients by Dirichlet sampling, leading to 30 sub-datasets with either non-i.i.d. features or non-i.i.d. categories. Under this setup, we average the text prompt tokens for clients in the same domain at the aggregation step. The results are summarized in Table 6. Compared to our default setting which each domain is considered as one client, our Fed-DPT only has 1.5% accuracy decrease when the dataset is further divided. In contrast, the conventional methods FedAvg and FedProx perform more sensitive to the non-i.i.d categories, with 3.6% and 2.9% accuracy decrease, respectively.

## 6 CONCLUSION

This work introduces Fed-DPT, a novel federated learning approach explicitly designed to address the key challenges of domain shift and communication efficiency. Our method strategically combines CLIP and prompt learning techniques for both visual and textual inputs, thereby enhancing parameter-efficiency and minimizing communication costs, while maintaining robustness in federated optimization involving heterogeneous data. Furthermore, we confront the pervasive issue of domain shift across clients by introducing domain-specific prompts and facilitating correlations between visual and textual representations through self-attention mechanisms. These innovations result in a domain-aware federated learning methodology that consistently demonstrates outstanding effectiveness. Notably, our experiments reveal a remarkable achievement—an average accuracy of 68.4% across six domains in the DomainNet dataset, marking an impressive 14.8% improvement over the original CLIP model. In comparisons with traditional federated learning methods like FedAvg and FedProx, as well as existing domain-agnostic CLIP-based approaches such as PromptFL and FedCLIP, our Fed-DPT consistently outperforms them across three benchmark scenarios.

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

APPENDIX

---

**Algorithm 1** Training Process of Fed-DPT

---

    **Input:**
        CLIP vision encoder $\boldsymbol{F}_V$, text encoder $\boldsymbol{F}_T$
        $n$ local datasets, each $\boldsymbol{D}_i = \{([\text{image}], [\text{class name}])_j\}_{j=1}^{J}$
        Total communication rounds $T$, momentum coefficient $\alpha$
    **Initialization:**
        Randomly initialize text prompts $[\boldsymbol{P}_T^1]^0, \ldots, [\boldsymbol{P}_T^n]^0$
        Randomly initialize visual prompts $[\boldsymbol{V}] = \{[\boldsymbol{v}]_1, \ldots, [\boldsymbol{v}]_n\}$
        Broadcast the pretrained model and prompts to $n$ clients
1:  **for** $t = 1$ to $T$ **do**
2:     *# Local training in parallel*
3:     **for** $i = 1$ to $n$ **do**
4:         Keep $\boldsymbol{F}_V$ and $\boldsymbol{F}_T$ frozen
5:         **for** $j = 1$ to $J$ **do**
6:             Compute $\boldsymbol{f}_T^k = \boldsymbol{F}_T(\boldsymbol{P}_T^k, [\text{class name}]_j)$ for $k \in \{1, \ldots, n\}$
7:             Compute $\boldsymbol{f}_V = \boldsymbol{F}_V([\text{cls}], [\boldsymbol{v}]_1, \ldots, [\boldsymbol{v}]_n, [\text{image}]_j)$
8:             Extract attention scores $\boldsymbol{w} = [w_1, \ldots, w_n]$ from $\boldsymbol{F}_V$ using Eq.5
9:             Weighted sum: $\boldsymbol{f}_T = \sum_{k=1}^{n} w_k \boldsymbol{f}_T^k$
10:           Compute $L_2$ loss: $\mathcal{L} = <\boldsymbol{f}_V, \boldsymbol{f}_T> / ||\boldsymbol{f}_V|| \cdot ||\boldsymbol{f}_T||$
11:           Update $[\boldsymbol{v}]_1, \ldots, [\boldsymbol{v}]_n$ and $\boldsymbol{P}_T^i$ by $\mathcal{L}$
12:           Update $\boldsymbol{P}_T^k, k \in \{1, \ldots, n\}, k \neq i$ by momentum: $\boldsymbol{P}_T^k = \alpha \boldsymbol{P}_T^k + (1 - \alpha)[\boldsymbol{P}_T^k]^{t-1}$
13:         **end for**
14:     **end for**
15:     *# Global aggregation in the server*
16:     Average $[\boldsymbol{V}] = \frac{1}{n} \sum_{k=1}^{n} [\boldsymbol{V}]^k$, where $[\boldsymbol{V}]^k = \{[\boldsymbol{v}]_1, \ldots, [\boldsymbol{v}]_n\}$ obtained from #$k$ client
17:     Assign $[\boldsymbol{P}_T^k]^t = \boldsymbol{P}_T^k$, where $\boldsymbol{P}_T^k$ obtained from #$k$ client
18:     Broadcast $[\boldsymbol{V}], [\boldsymbol{P}_T^k]^t (k \in \{1, \ldots, n\})$ to all clients
19: **end for**

---

**Discussion of Privacy.** As is shown in Table 5, similar to the CoOp results in its original paper, most decoded words do not have concrete meanings that can be simply interpreted, so the clients in our Fed-DPT cannot directly acquire the private domain information of others by these learnable prompts. In fact, the prompt learner generally tends to extract high-level and abstract textual features during the training process, rather than producing explicit labels such as "sketch" and "painting" that comply with human understanding.

Table 5: Nearest Words of textual prompts in DomainNet

| # | clipart | info g. | paint. | quick d. | real | sketch |
|---|---------|---------|--------|----------|------|--------|
| 1 | ˜ | fe | N/A | N/A | ° | kd |
| 2 | N/A | # | dng | , | ... | with |
| 3 | lh | bh | some | ? | N/A | N/A |
| 4 | and | N/A | lh | N/A | the | pjf |

**Fine-tuning performance**. Fine-tuning is a general method to transfer pretrained models to downstream tasks. However, fine-tuning the CLIP model may easily break its well-established vision-language alignment, which could undermine CLIP's strong capabilities in domain-generalization and open-vocabulary inference. So, in practice, the existing methods used to freeze CLIP's encoders and tune additional parameters such as prompt tokens and adapter layers from downstream tasks. An additional benefit of this protocol is its ability to produce favorable results without requiring a substantial volume of training data. In Table 6 (in the supplementary material), we obtain very competitive few-shot results by our prompt tuning technique. What's more, fine-tuning a large model such as CLIP entails a substantial increase in communication cost and decrease in convergence rate.

Table 6: Test accuracy (%) on DomainNet with 30 clients. Our results are marked in blue . The best results in each domain are **bolded**.

| Method | DomainNet | | | | | | |
|---|---|---|---|---|---|---|---|
| | clipart | infograph | painting | quickdraw | real | sketch | average |
| Zero-Shot CLIP | 66.1 | 40.6 | 62.3 | 13.5 | 80.4 | 58.5 | 53.6 |
| FedAvg | 37.6 | 56.4 | 55.6 | 31.0 | 71.9 | 57.2 | 51.6 |
| FedProx | 38.4 | 57.2 | 54.9 | 32.5 | 72.8 | 58.5 | 52.4 |
| PromptFL | 73.2 | 48.1 | 68.7 | 31.9 | 78.6 | 64.7 | 60.9 |
| FedCLIP | 72.7 | 47.0 | 66.2 | 32.8 | 76.9 | 57.2 | 58.8 |
| Fed-DPT (ours) | **75.8** | **62.3** | **69.0** | **39.5** | **83.9** | **70.6** | **66.9** |

Table 7: Comparison of Fine-tuning Performance in DomainNet

| Method | Backbone | Learnable params | acc. |
|---|---|---|---|
| FedAvg (McMahan et al., 2017) | CLIP ViT-B | 86M | 57.6 |
| FedProx (Li et al., 2020b) | CLIP ViT-B | 86M | 58.1 |
| Fed-DPT (ours) | CLIP | 16.9k | 68.4 |

So, given the same number of training iterations, the fine-tuning protocol often falls short to prompt learning. To provide a clear comparison, we have included the following results in Table 7.

Table 8: Few-shot accuracy (%) on DomainNet. $n$-shot denotes training with $n$ samples per class and per domain. Our results are marked in blue . The best results are **bolded**.

| Method | CLIP-based | full | 1-shot | 2-shot | 4-shot | 8-shot | 16-shot |
|---|---|---|---|---|---|---|---|
| Single Domain Tuning | ✓ | 59.1 | 51.1 | 51.8 | 53.2 | 54.7 | 56.2 |
| FedAvg (*ResNet-50*) | ✗ | 54.7 | - | - | - | - | 15.1 |
| FedAvg (*ViT-Base/16*) | ✗ | 55.2 | - | - | - | - | 19.7 |
| PromptFL | ✓ | 63.2 | 51.4 | 51.8 | 55.2 | 57.6 | 61.2 |
| FedCLIP | ✓ | 60.3 | 50.8 | 51.2 | 52.1 | 53.4 | 54.6 |
| Fed-DPT (ours) | ✓ | **68.4** | **55.4** | **57.2** | **60.3** | **62.7** | **64.5** |

**Robustness to few-shot learning.** One of the primary advantages of prompt learning is the robustness to few-shot scenarios. We investigate if our dual prompt tuning method retains this merit in the context of federated learning. Therefore, we conduct few-shot learning experiments on DomainNet, employing 1, 2, 4, 8, and 16 training samples per category and per domain. We evaluate the other CLIP-based methods with the same setting, yet only test 16-shot performance for FedAvg as it fails to yield reasonable results with fewer training samples. The corresponding results are summarized in Table 8. As is shown, CLIP-based methods exhibit superior robustness against few-shot learning than FedAvg, which again demonstrates the significant benefits of using parameter-efficient approaches. Also, our Fed-DPT consistently outperforms the baselines in few-shot learning.

**Convergence analysis**. Fed-DPT is optimized by applying a simple $L_2$ loss between the normalized visual and textual features. Formally, we have

$$L_2 Loss = -\text{sim}(f_v, f_t) = - < \frac{f_v}{||f_v||}, \frac{f_t}{||f_t||} >, \qquad (9)$$

where $f_v$, $f_t$ denote the representation vectors of the image and its corresponding text label, respectively. The general cross entropy loss can be written as

$$CrossEntropy = -\log \frac{\exp(\text{sim}(f_v, f_t)/\tau)}{\sum_{i=1}^{n} \exp(\text{sim}(f_v, f_t^i)/\tau)}. \qquad (10)$$

Since CLIP is a large-scale pretraining model that has established good vision-language alignment, the $L_2$ distance between an image feature $f_v$ and a mis-matched text feature $f_t^i$ tends to

be large and often exhibits low variance to different $f_t^i$. Consequently, the normalization term $\sum_{i=1}^{n} \exp(sim(f_v, f_t^i)/\tau)$ of the cross entropy loss tends to be a constant positive value, especially when the number of classes $n$ is big (e.g., 345 for DomianNet). So under this condition, minimizing the cross entropy loss is approximately equivalent to minimizing the $L_2$ loss $-d(f_v, f_t)$, and we personally find that leveraging this $L_2$ loss leads to slightly higher accuracy and faster convergence than cross entropy.

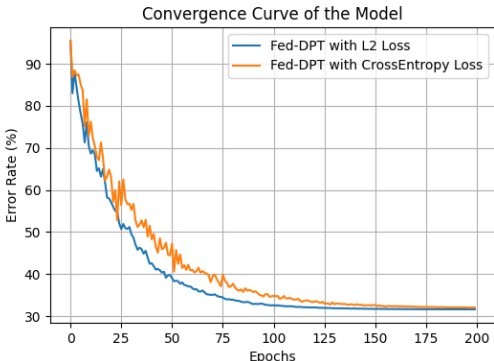

Figure 2: Comparison of convergence.

