# OpenReview forum: "Dual Prompt Tuning for Domain-Aware Federated Learning"
_ICLR.cc/2024/Conference — Submitted to ICLR 2024_

### Official Review · Reviewer_gJe5 · 2023-10-29

**Soundness:** 2 fair
**Presentation:** 3 good
**Contribution:** 2 fair
**Rating:** 5
**Confidence:** 4

**Summary:**

This paper presents a prompt tuning-based FL method that solves the domain shift problem in FL. Based on pre-trained vision-language model, both visual and textual prompt tuning strategies are utilized to facilitate domain adaptation. Experiments on CLIP model and benchmark datasets are conducted to show the performance enhancement achieved by Fed-DPT.

**Strengths:**

1. This paper has a unique contribution to the FL community by investigating the utilization of prompt tuning in dealing with domain shift problems.

2. A specific prompt tuning-based mechanism is developed under the FL framework and shows promising results compared with conventional FL methods and vision–language tuning FL methods.

**Weaknesses:**

1. It seems that the proposed Fed-DPT can only be applied to the vision-language model-based federated learning scenarios. Considering that there is a wide range of model architectures along with the domain shift problems, a more general method is preferred.

2. The novelty of textual prompt tuning and visual prompt tuning is limited. Since these two prompt tuning schemes were not first proposed by this paper, it is better to demonstrate the unique contributions of the prompt tuning part.

3. It is still implicit why the visual prompts can help detect the correlation between an input image and the domains and how this can help alleviate the domain shift problem.

4. It is better to also provide a comparison between the proposed method and traditional FL methods that address domain shift problems, such as FedBN[1] and some SOTA personalzied FL methods.

[1] FedBN: Federated Learning on Non-IID Features via Local Batch Normalization

**Questions:**

Will there be any privacy concerns when optimizing the prompts in an FL manner?

---

> ### Author Response · Authors · 2023-11-23
> **Response to Reviewer gJe5**
>
> **Q1: Fed-DPT is only applicable for vision-language models**
>
> **R1:** Thank you for your valuable comments. We agree that our method is specifically designed for vision-language models, but this does not mean that this study lacks contributions to the community. On the contrary, we believe that this work holds significant importance for research in federated learning.
>
> With recent advances in self-supervised and weakly supervised learning, pretraining a large-scale foundation model such as CLIP followed by lightweight prompt tuning for downstream tasks is becoming a new paradigm of transfer learning. Under this condition, the traditional fine-tuning-based protocols may not be as effective, and how to fully exploit the prompt learning techniques in federated learning has not been well explored. Our proposed Fed-DPT algorithm, which introduces coupled prompt learning at both vision and language ends, can effectively transfer the knowledge representations learned during the pre-training of CLIP models to downstream federated learning tasks without the need for excessive trainable parameters and communication overhead. We believe this study can offer valuable insights for the follow-up research of both vision-language models and federated learning.
>
> **Q2: Comparison to traditional FL methods**
>
> **R2:** Thank you for your valuable suggestion. In response, we have now included a comprehensive evaluation comparing our proposed method with traditional federated learning approaches, such as FedBN, in the revised version of our manuscript. Specifically, we replicated the FedBN setup on the DomainNet dataset, focusing on a 10-class subset of the original 345-class DomainNet to ensure a relevant and fair comparison. The results are summarized in the table below, where it shows that our method constantly outperforms the baselines.
>
> | Method         | clipart | info g. | paint. | quick d. | real | sketch | avg. |
> |----------------|---------|---------|--------|----------|------|--------|------|
> | FedBN  | 51.2    | 26.8    | 41.5   | 71.3     | 54.8 | 42.1   | 48.0 |
> | FedVPT    | 65.6    | 44.1    | 76.6   | 47.3     | 91.0 | 60.3   | 64.2 |
> | Fed-DPT (ours) | **83.6**  | **71.4**   | **87.2**  | **63.5**    | **96.8** | **79.2**  | **80.3** |
>
> **Q3: Effect of visual prompts**
>
> **R3:** Thanks for the comment. Regarding the concern you raised, we have provided a detailed explanation in Q2 of our overall rebuttal response.

---

### Official Review · Reviewer_bGQ3 · 2023-10-31

**Soundness:** 2 fair
**Presentation:** 2 fair
**Contribution:** 2 fair
**Rating:** 5
**Confidence:** 5

**Summary:**

Prompt learning is utilized to address the challenge of domain shift of training data between different clients. A novel approach called Federal Dual Prompt Tuning (Fed-DPT) is proposed, which uses pre-trained visual language models combined with text and image data. The experimental results demonstrate the effectiveness of the method.

**Strengths:**

1. Prompt learning is introduced into federated learning for solving the problem of domain transfer between clients.

2. The experimental results show that the method can improve the performance compared to the state-of-the-art methods under certain experimental settings.

**Weaknesses:**

1. The contributions need to be more clearly described. The combination of prompt learning and federated learning is a means of enhancing the effectiveness of the experiments. It lacks of innovation.

2. It is recommended to combine Figures 1 and 2, and draw a framework. And it’s better to list the whole algorithm. At present the overall process is not very clear.

3. At the end of subsection 4.2, it is mentioned that this paper does not observe the problem of training crash. It is just a summary of the observation from the experimental view, whether it is possible to make a theoretical analysis of the algorithm in terms of convergence or generalization bounds, etc.

**Questions:**

1. It is mentioned that some text information needs to be shared between clients such as class names, does it involve privacy protection?

2. In the section of experimental results, the performance is greatly improved on DomainNet dataset. It is suggested to analyze the reasons.

3.  The results of the ablation experiments show that the federated frameworks and textual cues are key factors for improving the effectiveness of the experiments. While the domain-aware mechanism slightly improves the performance. It’s better to give a more detailed reason.

4. Prompt learning is also adopted in "Efficient Model Personalization in Federated Learning via Client-Specific Prompt Generation" in ICCV2023. What is the difference between the proposed and the ICCV2023 methods, and it is suggested to add a comparison with this method in the experimental section.

---

> ### Author Response · Authors · 2023-11-23
> **Response to Reviewer bGQ3**
>
> **Q1: Comparison to pFedPG [ICCV'23]**
>
> **R1:**  Thank you for your valuable suggestion. The difference between our Fed-DPT and the pFedPG [ICCV'23] method lies in the model architectures and prompt learning mechanisms. Specifically, pFedPG focuses on single-modality (visual) prompt tuning with only a ViT-Base image encoder employed. In contrast, our Fed-DPT builds on top of a CLIP model, leveraging both visual and textual prompts. As we demonstrated, this dual-modality prompt tuning method facilitates the capabilities of detecting inter-domain correlations so it is suitable for domain-aware federated learning scenarios. Here we conduct a comprehensive evaluation using the experimental setup of pFedPG, which concentrates on a ten-class subset of the 345-class DomainNet dataset. The results are summarized below, where our Fed-DPT exhibits significantly superior performance than pFedPG.
>
> | Method            | clipart | info g. | paint. | quick d. | real | sketch | avg. |
> |-------------------|---------|---------|--------|----------|------|--------|------|
> | pFedPG | 73.0    | 50.1    | 84.3   | 60.0     | 94.0 | 68.4   | 71.6 |
> | Fed-DPT (ours)    | **83.6**  | **71.4**   | **87.2**  | **63.5**    | **96.8** | **79.2**  | **80.3**|
>
> **Q2: Clarifying the overall process.**
>
> **R2:** Thank you for your advice. We have combined Figures 1 and 2 in the revised manuscript to present a more clear framework. Further, to clarify the overall process, we include a pseudo-code of our Fed-DPT in the appendix. We hope these improvements in clarity can well address your concerns.
>
> **Q3: Chosen of L2 loss.**
>
> **R3:** Thank you for your constructive feedback. In this paper, we apply a simple L$_2$ loss between the normalized visual and textual features to optimize our model. Formally, we have
>
> $\text{L}_2\text{Loss} = -\text{sim}(f_v, f_t) = -<\frac{f_v}{||f_v||}, \frac{f_t}{||f_t||}>,$
>
> where $f_v$, and $f_t$ denote the representation vectors of the image and its corresponding text label, respectively. The general cross-entropy loss can be written as
>
> $\text{CrossEntropy} = -\log\frac{\exp(\text{sim}(f_v, f_t)/\tau)}{\sum_{i=1}^n\exp(\text{sim}(f_v, f_t^i)/\tau)}.$
>
> Since CLIP is a large-scale pretraining model that has established good vision-language alignment, the L2 distance between an image feature $f_v$ and a mismatched text feature $f_t^i$ tends to be large and often exhibits low variance to different $f_t^i$. Consequently, the normalization term $\sum_{i=1}^n\exp(\text{sim}(f_v, f_t^i)/\tau)$ of the cross entropy loss tends to be a constant positive value, especially when the number of classes $n$ is big (e.g., 345 for DomainNet). So under this condition, minimizing the cross entropy loss is approximately equivalent to minimizing the L$_2$ loss $-d(f_v,f_t)$, and we personally find that leveraging this L$_2$ loss leads to slightly higher accuracy and faster convergence than cross-entropy. We also added a figure (**Fig. 2**) in the appendix which summarizes the comparison of their convergence.
>
> **Q4: Performance improvements on DomainNet**
>
> **R4:** Thank you for your advice. Compared with OfficeHome and PACs datasets on which the Zero-Shot CLIP model already achieves high accuracy levels of 78.6\% and 95.8\%, DomainNet has a larger margin for improvements. As detailed in Table 1, our Fed-DPT reduces the performance variance among the six distinct domains in DomainNet, attaining the most significant improvements on the "information graph" and "quick draw" domains. This suggests that our domain-aware prompt learning method addresses CLIP's limitations in domain generalization, leading to significant robustness to image patterns. We have included more discussions on this point in the revised paper.
>
> **Q5: Significance of the domain-aware mechanism**
>
> **R5:** Thank you for your observation regarding the ablation study results. As indicated in Table 3, the Domain-Agnostic DPT, which employs both visual and textual prompts but does not incorporate our domain-aware mechanism, achieves a 9.9\% improvement over the baseline Zero-Shot CLIP. Our proposed Fed-DPT further enhances this by 4.9\%. This is a significant improvement in performance since the domain-aware mechanism does not require any additional model parameters compared with the Domain-Agnostic DPT approach. We will make it more clear in the revised version.

---

> ### Author Response · Authors · 2023-11-23
> **Additional explanation of Privacy**
>
> Thank you for your question about the privacy. In detail, during federated training of our method, there are class names as well as text prompts need to be shared across clients. In the context of federated learning studied in this work, the class names are not considered as private information since the clients are dealing with the same classification task so they all have the same class names. Also, sharing text prompt does not involve direct privacy leakage and we demonstrate that our method has the same level of privacy protection as FedAvg in our overall response. It is a good point to think over the privacy and we have added a detailed discussion for it in the revised paper.

---

### Official Review · Reviewer_WuFt · 2023-10-31

**Soundness:** 3 good
**Presentation:** 3 good
**Contribution:** 2 fair
**Rating:** 5
**Confidence:** 4

**Summary:**

This paper proposed Fed-DPT, a prompt learning-based technique to efficiently utilize the pre-trained vision-language model to mitigate the domain difference challenge in federated learning. Specifically, the participants of the FL system would locally optimize both language prompt and soft prompt for image encoding, and the server would aggregate these prompts and send them back to the clients. The experiments on several benchmark datasets show the performance of the Fed-DPT compared to the baselines.

**Strengths:**

1. The prompt tuning is lightweight and efficient for local training.

2. The target problem is novel and practical. Most of the FL literature are targeting on the non-iidness of the label distribution, but this paper concentrates on the domain difference of the local dataset, which is more challenging and practical in wild FL applications.

3. Compared to previous work such as PromptFL, Fed-DPT took both language and vision prompt into consideration.

**Weaknesses:**

1. For the textual prompt design and aggregation, would each client know the detailed information of the other's domain? As the setup in the experiment part, each client would only have one domain of images under extreme non-iidness, which means that the client would only know the domain name of its own. However, as each client needs to send its textual prompt to each other, each participant would know the detailed domain information about others, which is a privacy leakage.

2. As the paper concentrates on using the pre-trained vision-language model, the application scenario is limited to the cross-silo setups. The CLIP model is not practical to deploy in the cross-device FL setup.

**Questions:**

1. In the experiment part, why does the author not consider using the FedAvg/FedProx to directly fine-tune the CLIP model as the baselines?

---

> ### Author Response · Authors · 2023-11-23
> **Response to Reviewer WuFt**
>
> **Q1: Cross-silo vs. cross-device scenarios.**
>
> **R1:** Thanks for your valuable comments. We agree that the methods based on large-scale pre-training models are more suitable for cross-silo scenarios instead of cross-device setups. However, it is important to clarify that the two scenarios are characterized by the number of participants, the capability of computing resources, and the nature of the data involved, while many federated learning methods have a specific focus on one of them.
>
> Our method is basically designed to tackle domain shift challenges in federated learning, introducing a domain-aware prompt tuning mechanism and demonstrates effectiveness. In scenarios where each domain possesses substantial data with a distinct statistical distribution, our approach is particularly suited to a cross-silo federated learning context. We do not view this as a limitation; rather, it's a strategic focus. Solutions optimized for cross-device scenarios may not effectively address the unique challenges posed by cross-silo problems.
>
> We appreciate your suggestions to clarify the applicable scenarios of our method, and we have included this explanation in the revised version of the paper.
>
> **Q2: Fine-tuning CLIP with FedAvg/FedProx.**
>
> **R2:** Thank you for your constructive comments. In our study, we intentionally avoid fine-tuning the CLIP model due to a couple of significant considerations.
>
> Primarily, fine-tuning the CLIP model may easily break its well-established vision-language alignment, which could undermine CLIP's strong capabilities in domain generalization and open-vocabulary inference. So, in practice, the existing methods used to freeze CLIP's encoders and tune additional parameters such as prompt tokens and adapter layers from downstream tasks. An additional benefit of this protocol is its ability to produce favorable results without requiring a substantial volume of training data. In Table 8 (in the appendix), we obtain very competitive few-shot results by our prompt tuning technique.
>
> Secondly, fine-tuning a large model such as CLIP entails a substantial increase in communication cost and a decrease in convergence rate. So, given the same number of training iterations, the fine-tuning protocol often falls short of prompt learning. To provide a clear comparison, we have included the following results in our revised paper, which illustrates the advantages of our method.
>
> | Method            | Backbone   &nbsp;&nbsp; | Learnable params &nbsp;&nbsp; | acc. |
> |-------------------|-------------|------------------|------|
> | FedAvg | CLIP ViT-B | 86M              | 57.6 |
> | FedProx | CLIP ViT-B | 86M              | 58.1 |
> | Fed-DPT (ours)    | CLIP        | 16.9k            | 68.4 |

---

### Author Response · Authors · 2023-11-23
**Overall response to reviewers**

We sincerely appreciate all the reviewers for their valuable feedback and suggestions. Here is an overall response focusing on the questions most reviewers are concerned about.

**Q1: Concerns about privacy.** (For R#WuFt, R#bGQ3, and R#gJe5)

**R1:** In our method, every client acquires textual prompts from all others during the global aggregation phase. However, in prompt learning techniques, these prompt tokens are not considered as data but as a part of model parameters since they are learnable and updated by local gradients. Thus, our method with shared prompts has the same level of privacy-preserving capabilities as FedAvg which shares model parameters globally.

We also understand the concerns about privacy leakage of domain information by direct interpretation of the text prompts. However, it is actually very difficult to decode them back to meaningful words in natural language because these tokens are trained in a continuous space and their local optima are not necessarily associated with specific natural words. Here we follow CoOp~\citep{coop} to decode each text prompt by finding a standard vocabulary word with minimum Euclidean distance to it in the embedding space and summarize the interpretation results for DomainNet in the table below.

| #  &nbsp;&nbsp;| clipart &nbsp;&nbsp; | info g. &nbsp;&nbsp; | paint. &nbsp;&nbsp; | quick d.  &nbsp;&nbsp;| real &nbsp;&nbsp;| sketch &nbsp;&nbsp; |
|----|---------|---------|--------|----------|------|--------|
| 1 | ~       | fe      | N/A    | N/A      | °    | kd     |
| 2 | N/A     | #       | dng    | ,        | ...  | with   |
| 3 | lh      | bh      | some   | ?        | N/A  | N/A    |
| 4 | and     | N/A     | lh     | N/A      | the  | pjf    |

As is shown, similar to the CoOp results in its original paper, most decoded words do not have concrete meanings that can be simply interpreted, so the clients in our Fed-DPT cannot directly acquire the private domain information of others by these learnable prompts. In fact, the prompt learner generally tends to extract high-level and abstract textual features during the training process, rather than producing explicit labels such as "sketch" and "painting" that comply with human understanding.

We appreciate the reviewers for pointing out this concern. We have clarified it in the revised paper and will include this discussion in the final version.

**Q2: Novelty of dual-modality prompt tuning.** (For R#bGQ3 and R#gJe5)

**R2:** Thank you for your constructive comments. In response, it is important to clarify that our method is not a simple combination of federated learning with vision-and-language prompt tuning. Instead, the key contribution of this paper lies in the newly introduced domain-aware prompt learning mechanism. This significant advancement is evident in our ablation study (referenced in Table 3). By directly leveraging the multi-modality prompts for CLIP, the model reaches 63.5\% accuracy on DomainNet. However, our approach, which accounts for the correspondence between visual and textual prompts, achieves an additional 4.9\% improvement, raising the benchmark to 68.4\%.

Our design of domain-aware prompt learning is reflected in the weighted sum of $n$ text features by visual attention scores, where $n$ denotes the number of domains. This process can be considered as first classifying the domain of an image and then classifying its label. Specifically, for each task, we have $n$ visual prompt tokens and $n$ textual prompt sequences with one-to-one correspondence. When feeding the vision encoder with a [cls] token, $n$ prompt tokens, and a number of image tokens, the model can obtain the attention scores from [cls] to the $n$ prompt tokens and we consider these scores as the probabilities of the image belonging to the $n$ distinct domains respectively. Note that this is an intuitive process since higher attention scores indicate higher correlations between the image and the prompt token associated to a specific domain. On the language side, we encode each class name with $n$ distinct text prompts so that to obtain $n$ domain-specific representations of the class name. These $n$ text representations are then weighted averaged by Equation 6 in the paper, leading to the final target of the input image.

Therefore, in our Fed-DPT, the visual and textual prompts are strongly coupled where the domain correspondence is determined by self-attention scores in the vision encoder. We consider this streamlined design to be a key strength of our method. The domain-aware prompt learning mechanism, while not introducing any additional model parameters, still manages to secure a significant performance gain of 4.9\% on DomainNet.

---

### Meta-Review · Area_Chair_G6k6 · 2023-12-07

**Metareview:**

This paper considers a challenging federated learning setting, in which the local training data of each client originates from different domains. Specifically, the authors focus CLIP model and propose the dual prompt tuning technique to solve the domain heterogeneity issue in the federated learning setting. Preliminary experiments demonstrate the efficacy of the proposed approach.

Strengths:

(1)   The paper is well-written and easy to understand.

(2)   Applying prompt tuning techniques to solve heterogeneously federated learning is interesting.

Weaknesses:

(1)   The main concern of this work is about privacy leakage. Although the authors adopt the soft prompt techniques to get rid of privacy issues. However, the learned soft prompts are concatenated with tokens, which have a similar role to tokens and still have the risk of privacy leakage. In addition, recent works show that we can obtain the sensitive hard prompt soft prompts by projecting the soft prompt back to the hard prompt set.

[ref] Hard Prompts Made Easy: Gradient-Based Discrete Optimization for Prompt Tuning and Discovery, icml 2023.

(2)    Another concern is about the communication cost. When the number of clients is increasing, we should store all the VPT in the server and send them back to each client, which limits the scalability of the proposed framework.

(3)   The novelty of this proposed approach is limited. This work combines federated learning and vision-and-language prompt tuning techniques. Although the authors claim that the “domain-aware prompt learning mechanism” is the key contribution of this work, the explanation and technical contribution are not clear. In the future version, we recommend the authors put more effort into explaining the effectiveness of this part.

After the authors' response and discussion with reviewers, the concerns remain unresolved, and all reviewers recommend borderline rejection. Therefore, I recommend rejection.

**Justification For Why Not Higher Score:**

N/A

**Justification For Why Not Lower Score:**

N/A

---

### Decision · Program_Chairs · 2024-01-16

Reject